# Improving Monocular Facial Presentation–Attack–Detection Robustness with Synthetic Noise Augmentations

**DOI:** 10.3390/s23218914

**Published:** 2023-11-02

**Authors:** Ali Hassani, Jon Diedrich, Hafiz Malik

**Affiliations:** 1Information Systems, Security and Forensics Lab, University of Michigan-Dearborn, Dearborn, MI 48128, USA; 2Research and Advanced Engineering, Ford Motor Company, Dearborn, MI 48124, USA

**Keywords:** face, PAD, monocular, augmentation, texture, noise

## Abstract

We present a synthetic augmentation approach towards improving monocular face presentation–attack–detection (PAD) robustness to real-world noise additions. Face PAD algorithms secure authentication systems against spoofing attacks, such as pictures, videos, and 2D-inspired masks. Best-in-class PAD methods typically use 3D imagery, but these can be expensive. To reduce application cost, there is a growing field investigating monocular algorithms that detect facial artifacts. These approaches work well in laboratory conditions, but can be sensitive to the imaging environment (e.g., sensor noise, dynamic lighting, etc.). The ideal solution for noise robustness is training under all expected conditions; however, this is time consuming and expensive. Instead, we propose that physics-informed noise-augmentations can pragmatically achieve robustness. Our toolbox contains twelve sensor and lighting effect generators. We demonstrate that our toolbox generates more robust PAD features than popular augmentation methods in noisy test-evaluations. We also observe that the toolbox improves accuracy on clean test data, suggesting that it inherently helps discern spoof artifacts from imaging artifacts. We validate this hypothesis through an ablation study, where we remove liveliness pairs (e.g., live or spoof imagery only for participants) to identify how much real data can be replaced with synthetic augmentations. We demonstrate that using these noise augmentations allows us to achieve better test accuracy while only requiring 30% of participants to be fully imaged under all conditions. These findings indicate that synthetic noise augmentations are a great way to improve PAD, addressing noise robustness while simplifying data collection.

## 1. Introduction

One of the biggest challenges in robust face recognition (FR) is diverse data collection. For a reliable user experience, algorithms need to be trained on the target demographics across all application use-cases. This training is conducted on the entire (FR) pipeline: face detection [1,2], face identification [3,4] and presentation–attack–detection (PAD) [5,6,7]. PAD, in particular, introduces data complexity due to the adversarial nature of attack detection. The attacker’s goal is to identify a vulnerability in the algorithm, then expose that to gain unauthorized authentication [5]. This inherently increases the data collection needs proportionally to the number of realistic attack presentations—adding cost and development time. Hence, we believe that addressing FR training complexity is a pragmatic issue that needs to be addressed.

In this article, we focus on monocular PAD. State-of-the-art PAD methodologies traditionally involve dedicated 3D sensing [8]. These approaches are robust to all but the most sophisticated spoofing attacks, but necessarily introduce hardware costs. Conversely, re-using an existing camera for FR enables companies to offer more functionality without additional hardware costs. This is very appealing, where monocular FR is offered in phones [9], airports [10], buildings [11], and even cars [12].

Monocular PAD methods are often based on fine artifact detection methods [6,7,13]. Spoofing articles are rarely 100% perfect representations of the real face, where artifacts can be observed in textural aberrations [14], facial “correctness” [15], and light reflectance [16]. These features work well in measured laboratory conditions but can be impacted by the imaging environment. Common imaging noises, such as poor focus, under/over exposure directly and lighting artifacts can disrupt fine artifact detectors. These noises can present in everyday use, but are often under-represented in open-source FR datasets [6,17,18,19].

We visualize our PAD training, which is particularly complex in Figure 1. Here, we represent a subset of the common collection use-cases, which need to be repeated across demographic representatives. One can imagine how attack presentations can be highly variant, and therefore require very sensitive detection algorithms. This can be partially mitigated by picking the right imaging spectrum. Our figure juxtaposes the human perspective (RGB camera) against the near-infrared, where it is obvious that the contrast between live faces and spoofs increases due to differences in material reflectance [16]. Still, there are concerns with over-fitting. Imaging noise can overlap while artifacts are introduced by the spoof presentations, introducing detection errors.

Data augmentation is a popular means to address data collection complexity. In particular, geometric and contrast-based methods are rather popular. Images can be flipped, rotated, zoomed, cropped, masked, and contrast-degraded to vary perspectives [20,21]. Our evaluations show that these improve features, but are insufficient for sensor and lighting noises. We address this with a toolbox of physics-informed noise generators, introducing sensor and lighting artifacts. We validate the toolbox by first doing a noise-robustness evaluation, benchmarking against popular augmentations on noisy images. Our findings show that the toolbox not only generalizes the best on noisy data but actually improves accuracy on clean data. We hypothesize that this is because including realistic noises in training helps better discern spoofing artifacts from sensor or lighting artifacts. We verify this hypothesis in an ablation study, demonstrating that we can use noise augmentations as a means to simplify data collection. This is achieved without the need of complex 3D modelling for traditional lighting simulations [22]. We summarize contributions as follows:Developing a physics-informed noise-augmentation toolbox with source-code: https://github.com/AHassani92/Face-Perception, accessed on 1 June 2023.Validation toolbox improves PAD noise robustness over popular augmentations;Validation toolbox can replace real PAD data by generating highly robust features.

We present the rest of the paper as follows: the literature survey is in Section 2, the proposed augmentation methodology is in Section 3, the experiment design is in Section 4, experiment results are in Section 5, and conclusions are in Section 6.

## 2. Related Works

Face presentation–attack–detection (PAD) algorithms secure authentication systems against spoofing attacks. Spoofing attacks are typically derived from an enrolled user’s image (e.g., acquired from social media) and projected onto a facsimile. Typically, facsimiles are pictures, videos, paper-masks, and fabric masks due to cost and construction complexity, though they can also include 3D latex or silicone masks [5].

We start this literature survey by introducing the general PAD methods, starting with depth and motion. Next, we review state-of-the-art fine artifact detection. These are the most likely to be used today’s applications and are the emphasis of the survey. We discuss them and the associated noise robustness, noting that image quality metrics can mitigate the problem by filtering out imagery that will likely fail [23]. Next, we review the relevant augmentation techniques. We lastly discuss these findings in the context of pragmatically addressing noise robustness.

### 2.1. Presentation Attack Detection General Methodologies

We view depth sensing as the go-to PAD technology. Simple spoofs lack realistic 3D-geometry and are easily detectable with depth. These methods originate with stereo-cameras [24], calculating the disparity between images. More modern approaches use either time-of-flight lasers [25] or triangulated light emission [26], where structured light triangulation improves depth precision to millimeters. Many of the best competition works utilize these methods [13]; again, the issue is cost. For a PAD technology to achieve mass-adoption, it should be compatible with existing imaging systems.

Alternatively, we observe that motion methods are also popular. These approaches pursue temporal patterns in facial structure, where spoofs are often constrained [27]. The simplest approach is typically blink-rate, tracking eye-landmarks [28]. Heart-rate detection can also be employed to track blood flow pulsing across the face [29]. Periodic behavior is measured across the forehead [29], often isolating the green-channel (due to hemoglobin absorption) [30]. That said, these motion approaches have fundamental drawbacks. In many cases, a video will defeat all of them [31]. Fabricated masks can also expose eye holes or forehead patches (to present eye and/or heartbeat motion). It is also essential to note that many methods are sensitive to scene movement, and can require at least 30 s of video to achieve a stable signal [30].

### 2.2. Presentation Attack Detection Fine Artifact Methodologies

We find that monocular methods typically pursue attack instrument production artifacts [5]. Common examples are face-distortion and color-quantization during printing [32]. Furthermore, the spoof material and geometry may not interact with light in the same way that as live faces. This can bias the distribution and therefore the perceived texture [32].

In principle, texture-artifact methods should be well-suited for PAD. Wen et al. demonstrate that color distribution analysis can identify reproduction artifacts [33]. This work, however, is proven out in static lighting conditions. Others have been unsuccessfully able to reproduce the results with dynamic lighting (including our own lab [16]). Chingovska et al. alternatively use a popular texture descriptor, the local-binary-pattern (LBP) [34], noting it cannot discern liveliness for RGB cameras. We similarly concluded that LBP is sensitive to visible light variance, but robustness can be notably improved when using the illuminated near-infrared [16].

We note that deep-learning approaches can notably improve performance. Several relevant competition winning algorithms are the central difference convolutional network (CDCN), generalized spoof-cues network (GSCN) and dual-branch depth network (DBBN). The CDCN employs a specialized layer (i.e., the central difference convolution) to more precisely discern textural artifacts from noise, then infers a binary liveliness-map [14]. This network won the CVPR 2020 facial anti-spoofing challenge [14]. The GSCN alternatively generates a spoof “cue” map (i.e., what a spoof should look like) for comparison against the input [15]. This network won first place on the FaceForensic deepfake challenge [15]. Lastly, the DBBN employs meta-learning to correlate liveliness and depth-estimation features [35]. This network has excellent cross-dataset generalization, and is validated by TIFS [35]. All three of these are benchmarked in our prior work. While they do excellently in competition data, they actually over-fit when introducing a high-variance spoofing dataset (and are outperformed by a simple MobileNet binary classifier) [16]. This is the fundamental challenge of PAD. When pursuing fine artifacts, there is potential for false-positives with imaging noise. This highlights the need for a noise-robust feature space.

We observe that others propose that combining texture and motion can improve generalization, e.g., the best of both worlds. The simple approach is to sequentially aggregate texture features, e.g., generating optical-flow maps from a sequence of LBP descriptors [36]. These approaches are limited, however, as they do not directly integrate temporal and texture features. Spatio-temporal networks, alternatively, are designed exactly for this purpose [37,38]. Spatio-temporal networks do in fact improve accuracy, but have the drawback of notable computational complexity. This re-iterates the concern with depth sensing: mass-scale adoption. One potential compromise between accuracy and computation is to employ simpler networks with a combination of image-quality assessment filters [23]. These filters can identify images not suitable for biometric authentication [23]. Note, however, that there is also the risk of undesirably denying service. For example, phones that use selfie-cameras for FR do not work in low-light and require the user to authenticate with alternative means [9].

### 2.3. Augmentation Methodologies

We pose a simple question: if popular monocular methods are sensitive to noise, can we gain robustness through augmentations? Our survey shows there are a number of popular augmentation strategies in the face recognition space. The simplest are geometric transformations: image flipping, rotating and cropping, padding, etc. [20]. These techniques are easy to implement and consistently demonstrate benefits. Some more sophisticated approaches is to introduce synthetic variance, such as adjusting hairstyles, glasses and pose [20]. These techniques are more useful when trying to address identification bias, though they do not pertain to addressing sensor or environmental artifacts. This approach is followed up by others who use employ full-synthetic rendering techniques to generate specific demographic targets (e.g., gender, ethnicity, accessories) [39,40]. These approaches can help fill in gaps in the collection space (again, useful for addressing identification bias) but are not designed for system noise. Furthermore, fully synthetic training datasets do not generalize as well as actual imagery [40].

We believe that an ideal solution is to instead synthetically augment actual imagery with noise. Uchoa et al. propose simulating illumination effects by adjusting the brightness, saturation and contrast of the images [41]. Crispell et al. alternatively present arguably a powerful approach of projecting faces onto a 3D model, then synthetically adjust the pose and illumination [22]. Moreno-Barea et al. more specifically target imaging noise, and generate training data with variable noise [42]. Pervaiz similarly find that introducing noise when training speech-recognition algorithms [43]. We believe these approaches show there is promise in pursuing synthetic noise augmentations, and suggest opportunity in pursuing sensor and environmental augmentations.

### 2.4. Addressing Noise Robustness in Pragmatic Fashion

Addressing noise robustness by collecting data under all expected conditions is expensive and unrealistic. Instead, synthetic noise augmentations can be an ideal middle-ground. Real-world authentication can involve a large variety of sensor and environmental effects, where we have prior established that even state-of-the-art algorithms need a strong signal-to-noise ratio [16]. Hence, we build off the works by Crispell, Moreno-Barea, and Pervaiz and present a physics-informed noise-augmentation toolbox. We present the generators’ methodologies and validate their utility in training in the next sections.

## 3. Physics-Informed Noise-Augmentations Methodology

We propose that face presentation–attack–detection (PAD) robustness can be improved with physics-informed noise generators. Real-world scenarios can introduce noises that affect image quality, degrading sharpness and contrast. We believe relevant scenarios can be modeled through physics-informed generators, and are synthetically applied in lieu of actual data collection. Recall from the related works section that that synthetically injecting noise has shown promise in other applications, including speech recognition. We build off this approach and introduce generators that pragmatically address imaging sensor and environmental (lighting) noises. We select these because they are known to commonly present and have well known physical effects.

We visualize the noise augmentation types in Figure 2. There are two fundamental categories: sensor (designated by ${}^{1}$) and environmental (designated by ${}^{2}$). The goal is to use physics-informed techniques to make semi-realistic augmentations. These augmentations perturb the feature space in a relevant fashion, but are not necessarily photo-realistic. Ideally, the augmentations would in fact be photo-realistic and perfectly capture lighting effects. However, this traditionally requires precise 3D modelling [22]. Three-dimensional modelling is often not available for monocular imaging (e.g., would require additional depth imaging or simulation) and is computationally complex. This provides the benefits of being cost effective and highly scalable.

### 3.1. Synthetic Noise Generators

We select these noises categories because of they are realistic to present in real-world scenarios. For example, it is possible for images to corrupt in storage or memory, but these are failures with the hardware that are independent of the software. All sensors, however, are likely to have some noise caused by the sensor and auto-exposure algorithms. It is inevitable that the sun will eventually introduce bright spots and shadows on the user’s face. These noises are extremely difficult to filter out with hardware. Given their frequency of occurrence, it seems logical to address algorithm robustness to them.

Furthermore, all of these noises are well understood from an optics perspective. There are well known mathematical models that can be simulated with open-source computer vision tools [44,45]. We detail the noise generators next, given the generators and the methodology in Table 1. All source code can be accessed at the following github: https://github.com/AHassani92/Face-Perception, accessed on 1 June 2023.

#### 3.1.1. Camera Noise: Blur

Camera focus is required to ensure a crisp facial-image. A lens being out of focus consequently then results in blurry imagery [46]. This is simulated using a Gaussian blurring kernel. This noise generator is implemented utilizing the Science-Kit Image toolbox with the Gaussian filter tool [45].

#### 3.1.2. Camera Noise: Dark-Current

Camera photo-receptors are imperfect, and can leak current even when no light is supplied [47]. This effect is essentially randomly supplying pixel intensities, and can be simulated using a Gaussian distribution. This noise generator is implemented utilizing the Science-Kit Image toolbox with the Gaussian distribution tool [45].

#### 3.1.3. Camera Noise: Shot

Light is really never perfectly uniform. The photons are often received in a stochastic process, which is defined as shot noise [48]. This effect can be modelled by using a Poisson process, effectively adding a “pepper” effect to the image. This noise generator is implemented utilizing the Science-Kit Image toolbox with the Poisson tool [45].

#### 3.1.4. Camera Noise: Salt and Pepper

Analog-to-digital conversion is a key part of translating photo-receptors into a image-processing format. At times, this process can erroneously result in random saturation (white pixels) or lost signal (black pixels) [49]. This noise generator is implemented utilizing the Science-Kit Image toolbox with the salt and pepper tool [45].

#### 3.1.5. Camera Noise: Under-Exposure

Cameras typically expose light until a basic set of contrast metrics are met [50]. These metrics are often a simple count of white and black pixels, and as such the overall image can be under-exposed if there are notable bright spots in the scene [50]. This causes the exposure-time to become biased incorrectly to be too short, resulting in the face being very dark (non-coincidentally also presenting dark-current noise). This effect can be modelled by adjusting the gamma to be darker such that facial features start to disappear. This noise generator is implemented utilizing the Science-Kit Image toolbox [45].

#### 3.1.6. Camera Noise: Over-Exposure

Over-exposure is the opposite problem of under-exposure. Due to dark spots in the image, the overall exposure time is increased to be too high. This results in the face looking saturated [50]. This effect can be modelled by adjusting the gamma to be brighter such that facial features start to disappear. This noise generator is implemented utilizing the Science-Kit Image toolbox [45].

#### 3.1.7. Environment Noise: Point-Source

Point sources present in a point-like fashion on the face [51]. This results in the region being particularly bright, often presenting shot noise (with non-source region consequently under-exposed). This is simulated using a randomized ellipse using OpenCV v4.1 [44]. The region within the ellipse is then over-exposed and the region outside the ellipse is under-exposed (using the generators proposed). The boundary between regions is blurred.

#### 3.1.8. Environment Noise: Point-Shadow

Point shadows are the inverse of a point source [51]. This results in the shadow region being particularly dark, often presenting dark-current noise (with the non-shadow region consequently over-exposed). This is simulated using a randomized ellipse using OpenCV [44]. The region within the ellipse is then under-exposed and the region outside the ellipse is over-exposed (using the generators proposed). The boundary between regions is blurred.

#### 3.1.9. Environment Noise: Streaking-Source

In some cases, a specular source may present itself overhead the user. If they have an obstruction, such as wearing a hat or the roof of a vehicle, anecdotal evidence shows that this results in the bottom half of the face being illuminated by a bright streak (with the non-streak region being consequently under-exposed). This is simulated using a randomized streak using OpenCV [44]. The region within the streak is then over-exposed and the non-streak region is under-exposed (using the generators proposed). This boundary is blurred.

#### 3.1.10. Environment Noise: Streaking-Shadow

Opposite to a light-streak, the specular source may present below the camera. Anecdotally, this illuminates the face top-half with a bright streak (with the non-streak region consequently under-exposed). This is simulated using a randomized streak using OpenCV [44]. The region within the streak is then over-exposed and the non-streak region is under-exposed (using the generators proposed). This boundary is blurred.

#### 3.1.11. Environment Noise: Piping Source

Alternatively, the specular source may present at an angle to the user. This can happen when the user is facing north or south, and the sun is oriented to the east or west (depending on time of day). Anecdotal evidence shows that this creates a bright, light-pipe across the user’s face (with the non-pipe region consequently under-exposed). This is simulated using a randomized pipe across the face using OpenCV [44]. The region within the pipe is then over-exposed, and the non-pipe regions are under-exposed (using the generators proposed). The boundary between regions is blurred.

#### 3.1.12. Environment Noise: Piping Shadow

Lastly, a specular source can be obstructed by a large object that casts a piping shadow. This can occur when the user is underneath a large structure, such as driving under a bridge. Anecdotal evidence shows that this creates a dark, shadow pipe across the user’s face (with the non-pipe region consequently over-exposed). This is simulated using a randomized pipe across the face using OpenCV [44]. The region within the pipe is then under-exposed and the non-pipe regions are over-exposed (using the generators proposed). The boundary between regions is blurred.

## 4. Presentation–Attack—Detection Experiment

Our experiment validates the utility noise-augmentations on real-world generalization. In the past, we have collected liveliness data in controlled laboratory conditions to demonstrate how near-infrared reflectance is a robust PAD feature [16]. Here, we expand our PAD dataset to introduce realistic sensor and lighting noises that would intentionally degrade the PAD features. We hypothesize that our augmentation toolbox will improve robustness by inherently teaching the algorithm where image components are associated with PAD and which are associated with noise. We begin this process by recapping the original data collection, then describing the two conducted experiments.

### 4.1. Face Presentation–Attack—Detection Dataset

Our baseline evaluation dataset is a robust collection of different people and perspective variance. In the prior paper, we imaged a diverse array of 30 adults. We considered gender (20 males and 10 females), ethnicity (6 groups) and age (6 groups). Each participant performed a script, where they were coached to act out common behaviors that vary in head-pose and distance to camera. Behavioral script examples included sitting in a car, adjusting the mirrors, interacting with the infotainment system, and having a conversation. Again, these were designed to be typical behaviors that introduce the facial presentation variance.

They performed the behavioral script as a live person and while presenting four spoof-instruments. Everything was repeated under three lighting conditions: laboratory-dark (940 nm illumination only), laboratory-light (940 nm illumination with all lab lights turned on) and diffuse outdoor sun (940 nm illumination with sunlight diffused by glass). We give the collection scenarios in Table 2.

We show dataset face-crop samples in Figure 1. This visualizes how imaging a complete matrix of scenarios is robust but complex. This generates approximately 80,000 unique frames for PAD evaluation. Imaging is conducted with a 5 mega-pixel FLIR Blackfly monochrome camera [52] employing a 940 nm filter with matching illumination. It is inherently a very repetitive process; hence, part of the experimentation is to identify where collection can be simplified in an ablative fashion. Note that the RGB samples in Figure 1 are only shown to visualize the human perspective. All experimentation (training and testing) is conducted using the illuminated near-infrared data.

### 4.2. Noise Robustness Evaluation

We perform a pair of experiments to demonstrate that our augmentation toolbox generates better features than other popular methods. For simplicity, we select the top deep-learning algorithm (MobileNetV3 binary classifier) from our previous paper [16]. This algorithm is known to be robust when acting on noise-controlled data. Our evaluation is then conducted in two phases: first in a sensitivity analysis, then in a data ablation study.

In all scenarios, we use 70% of the participants for training, 20% for validation and 10% for testing. We pre-process all images with the Retina Face network [2], cropping the face and then re-scaling to the appropriate deep-learning network input size. The standard PAD evaluation metrics are used: average-classification-error-rate (ACER), non-presentation-classification-error-rate (NPCER, essentially FRR) and attack-presentation-classification-error-rate (APCER, essentially FAR). Also note that traditional consumer displays do not emit in near-infrared [53]. This means the display-replay attacks are not present to the near-infrared camera, a passive form of spoof rejection. These images without a detected spoof-face are not reflected in the algorithm validation metrics.

### 4.3. Augmentations for Benchmarking

We introduce two popular augmentation techniques as evaluation controls. The first is the geometric approach. This includes image translations, flips, rotations, randomizing cropping and randomized padding [20]. The second is the brightness, saturation and contrast (BSC) approach. The BSC is a wellknown toolbox that adjusts image properties associated with brightness, saturation and contrast (hence the name) [41].

#### 4.3.1. Exp. 1: PAD Algorithm Noise Sensitivity

Our first question is whether algorithms learn to differentiate liveliness features from sensor and environmental noises. Recall that our PAD methodology is based on detecting fine-artifacts in the imagery, and therefore is particularly sensitive to image artifacts.

We do this by generating a noisy dataset with the proposed augmentation toolbox. We next evaluate five training approaches on the original dataset and the noise-augmented dataset. These training conditions are as follows:Original dataset;Original dataset augmented with geometric methods [20];Original dataset augmented with brightness, saturation and contrast method [41];Original dataset augmented with proposed toolbox;Original dataset and original augmented with proposed toolbox.

#### 4.3.2. Exp. 2: Noise-Augmentation as Data Replacement Ablation Study

We hypothesize that if noise-augmentations generate better features, this technique can also be used to replace actual data collection. The prepared dataset is fully contrastive, i.e., all participants perform all activities. This is robust, but very time consuming. Being able to achieve similar classification accuracy with simplified collection would both reduce time and cost associated.

We evaluate this by next ablating the training datasets. The test datasets are still complete (i.e., fully-contrastive), where we now introduce new training subsets based on liveliness-pairs. A liveliness pair is defined and these subsets are as follows:Fully-contrastive: all participants have both live and spoof imagery;Partially-contrastive: 30% of participants have both live and spoof imagery;Not-contrastive: no participants have both live and spoof imagery.

### 4.4. Research Limitations

Facial attack presentations constantly evolve based on the detection methods. Each PAD method inherently will have vulnerabilities; it is only a question of the complexity of the reproduction [5]. Hence, this experimentation is designed to capture relevant attacks and imaging use-cases, but is by definition not all-inclusive. Furthermore, the test dataset is also made of synthetic noises, which are semi-realistic but not real data. Ideally, we would prefer to repeat the initial experiment with the same participants; they are not available. Lastly, it is important to acknowledge that all of these scenarios are inherently constrained because the data are fixed. This is a pragmatic approach, but a best practice would also involve independent penetration testing.

## 5. Experimentation Results

Our evaluation goal is to demonstrate the value of using noise-augmentations in training. Here, we show that the selected PAD algorithm is by default sensitive to noise (as expected), but becomes robust using the training augmentations. We also show that our noise-augmentations help inherently find better features in a training data ablation study. This suggests that noise augmentations can be used in lieu of actual data, simplifying the collection process. We present these results with discussion next.

### 5.1. Exp. 1: PAD Algorithm Noise Sensitivity

Our first experiment is an augmentation sensitivity analysis. We do observe that there is a general utility to augmentations. Training with both the geometric and BSC augmentations improve classification error-rates on the original dataset, with the BSC being incrementally better. Intuitively, this make sense. However, all three controls (original data and two comparison augmentations) fail to generalize on sensor and environmental noise presentations. This is clear from the ten-fold increase in error rates when testing on noisy-data. We can robustly address this, however, by including noise-augmentations in training. We provide these results in Table 3.

Including the proposed noise-augmentations in the training dataset directly resolves generalization on noisy data. This result is rather intuitive, and a simple validation that including noise in training data is necessary for fine artifact detection PAD algorithms. Perhaps more interesting, is we actually observe accuracy improvements on the original data when training with noise augmentations. This occurs both when including noise-augmented data with the original training dataset, and when training on noise-augmented data alone. We hypothesize that this is because adding many noisy variants of each image enables the algorithm to differentiate image-components associated with liveliness vs. those with noise. This inherently does a better job optimizing the PAD features. This suggests that noise-augmentations have value regardless of operational conditions.

### 5.2. Exp. 2: Noise-Augmentations as Data Replacement

We hypothesize in Exp 1 that training with high-variance noise can inherently de-noise PAD features. This is inspired from the noise-augmented training data actually generalizing better on original data than training on original data alone. Here, we demonstrate that this feature robustness can actually be used as a means to reduce data-collection complexity. We provide these results in Table 4.

We demonstrate that partially contrastive data can achieve an arguably better performance when using noise-augmentations. By default, there is a notable penalty for not using fully contrastive data. Observe that the rows of partially contrastive and non-contrastive training sets show notable drops in test accuracy. This is to be expected, as the datasets are by design degraded. The addition of noise augmentations, however, improves performance in all cases. Note that now using noise augmentations on partially contrastive data (the ‡ rows) actually outperforms the original dataset on original test data (the * rows). Furthermore, the performance is negligibly worse than the best condition of fully contrastive data (the † rows).

This is a very useful finding. It demonstrates that noise augmentations enable simplifying data-collection to be partially contrastive, with minimal risk to feature generalization. To appreciate the value, consider the ramifications of addressing a newly discovered vulnerability. Each time a new attack is discovered, developers traditionally need to do a comprehensive collection across many participants and scenarios with the new attack. With this noise augmentation methodology, we can justify only doing a partially contrastive update with the new attack and expect to achieve similar results.

## 6. Discussion

We demonstrate in this paper that synthetic noise augmentations can improve real-world presentation–attack–detection (PAD) robustness. We present a toolbox of twelve noise augmentations that introduce sensor and environmental noise in a physics-informed fashion. We then evaluate these noise augmentations against popular augmentation techniques to determine which approach generates the best features. Our experimental results demonstrate that our approach not only generalizes better on noisy-data, but can help reduce data collection complexity.

First, we do a noise sensitivity analysis to benchmark the augmentation technique. While the original training data with popular augmentations fail to generalize well on noisy data, training with our proposed approach successfully retains robustness. Classification accuracy actually improves even on clean data. This suggests that training with the noise-augmentations in general results in better features. We believe that this is because the noise-augmentations perturb the feature space in a highly variant yet realistic fashion, such that the algorithm is inherently able to differentiate PAD features from noise.

Building on this observation, we hypothesize that noise augmentations can be used as a replacement for real data. Collecting the full matrix of scenarios is ideal, but very complex. To evaluate this, we perform an ablation study that removes liveliness pairs, e.g., participants may either have live or spoof data, but not both. This introduces two new degraded training datasets: partially contrastive and not contrastive. This second evaluation shows that using partially contrastive data with noise augmentations can actually generate better features (as measured on clean data) than training on the original dataset by itself.

This is a powerful finding. Generating better features in this approach suggests PAD algorithms can be optimized by having some participants perform the complete set of imaging procedures, then targeting specific scenarios with the others. This is not only a cost-save but allows for quickly adapting as new attacks are discovered with a partially contrastive update.

In conclusion, we believe that noise augmentations are a valuable tool for gaining real-world robustness. Regardless of authentication technology, noise is ever-present. This approach not only addresses noise robustness, but improves features on clean data and reduces data-collection complexity. Next steps are centered around expanding the use of noise. One proposal is to repeat this evaluation while imaging the actual participant in noisy conditions. This improves the integrity of the test condition and better measures feature generalization. This was originally intended, but not possible due to the original collection participants not being available. Additionally, we propose that noise-presence can be classified in a meta-learning paradigm. We have demonstrated in other works that facial context can improve identification accuracy over pose [54]. Here, the idea would be to classify what noises are present to better differentiate liveliness features from noise. Lastly, we propose comparing the capability to contextualize noise in the meta-learning network vs. vision transformers, which include attention layers by design.

## 7. Patents

This research has generated patent applications jointly filed between the Ford Motor Company and the University of Michigan. If allowed, a patent number is provided; those that are still in process are identified by case ID.

COUNTERFEIT IMAGE DETECTION (USPTO Case ID: 84238879US01). Convenience security facial authentication using near-infrared camera specular reflectance. The person is first identified; then, it is verified that their compensated specular reflectance meets the liveliness-enrollment-similarity score.COUNTERFEIT IMAGE DETECTION (USPTO Case ID: 84227552US01). It is a secure facial-authentication capable of detecting complex 3D masks via co-registered CMOS and thermal cameras. CMOS camera is used to detect and identify the face; liveliness is determined using thermal analysis. The system is secure with very efficient liveliness analysis.MATERIAL SPECTROSCOPY (USPTO Case ID: 84279449US01). Material source-identification using combined RGB-IR spectroscopy analysis. RGB provides material color context for near-infrared material-spectroscopy. This provides a naive Anti-Spoofing approach (vs. specular-reflectance verification against enrollment).MATERIAL SPECTROSCOPY (USPTO Case ID: 84279422US01). Facial optical-tethering methods for material-spectroscopy liveliness-analysis. Facial distance and orientation are determined using deterministic key-points or deep-learning.MATERIAL SPECTROSCOPY (USPTO Case ID: 84279413US01). Facial environment-compensation methods for material-spectroscopy liveliness-analysis. Sequenced light toggling is used to detect the face with an illuminated frame and de-noise the background using a non-illuminated frame analysis.MATERIAL SPECTROSCOPY (USPTO Case ID: 84279409US01). Facial segmentation methods for material-spectroscopy liveliness-analysis. In particular, the emphasis is placed upon segmenting “skin” pixels, either using deterministic key-points or semantically using deep-learning.

## Figures and Tables

**Figure 1 sensors-23-08914-f001:**
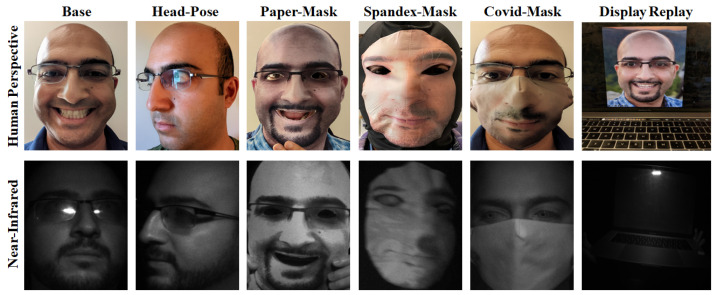
Presentation–attack–detection data collection visualization. Quality datasets include live and spoof presentations over many scenarios, such as pose and distance. We show both the human and near-infrared perspectives to illustrate how picking the right spectrum of light can help better discern artifacts due to spoofing vs. imaging noise.

**Figure 2 sensors-23-08914-f002:**
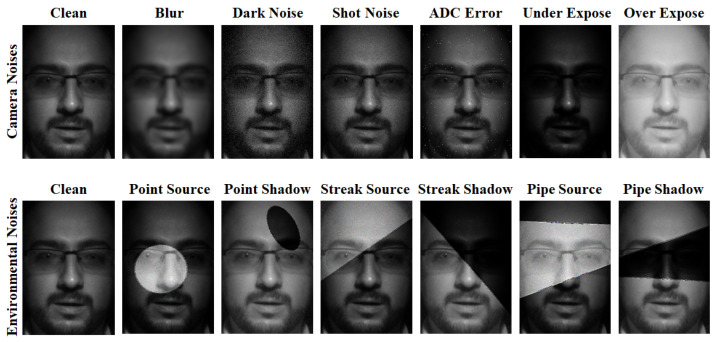
Visualizing physics-informed noise-augmentations. Observe how the camera and environmental noises are designed to perturb the algorithm in the same fashion that real-world noises do without complexity of being photo-realistic.

**Table 1 sensors-23-08914-t001:** Synthetic camera and environmental noise augmentation generators. This table enumerates both the relevant types of noises and how they are being generated for evaluation. Camera noises are indicated by ${}^{1}$. Environmental noises are indicated by ${}^{2}$.

Noise	Generator
Camera blur ${}^{1}$ (poor focus)	Gaussian Low Pass Filter
Dark noise ${}^{1}$ (random leakage)	Gaussian Noise Generator
Shot noise ${}^{1}$ (random photo distribution)	Poisson Noise Generator
Salt and pepper noise ${}^{1}$ (analog-to-digital error)	Random 0 and 255 Generator
Under-exposure ${}^{1}$ (low contrast)	Gamma Subtraction
Over-exposure ${}^{1}$ (saturation)	Gamma Addition
Point-source ${}^{2}$ (point sources)	Synthetic Bright Ellipse
Point-shadow ${}^{2}$ (shadows)	Synthetic Dark Ellipse
Streaking-source ${}^{2}$	Synthetic Overhead Sun
Streaking-shadow ${}^{2}$	Synthetic Overhead Shadow
Piping-source ${}^{2}$	Synthetic Side Sun
Piping-shadow ${}^{2}$	Synthetic Side Shadow

**Table 2 sensors-23-08914-t002:** Liveliness experiment data collection matrix. All participants perform all behavioral presentations, with the exception of sun-diffuse (all spoof with some live, indicated by *). Note that participants are coached to make common place behaviors to generate the varying perspectives in a natural fashion (vs. attempting specific distances and head-poses).

Presentation	Lighting	Position	Yaw	Pitch
		**(Meters)**	**(Deg)**	**(Deg)**
Live (30)	Lab-Dark, Lab-Lights, Sun *	[0.5, 1.5]	[−45, 45]	[−15, 15]
Display-Replay (30)	Lab-Dark, Lab-Lights, Sun *	[0.5, 1.5]	[−45, 45]	[−15, 15]
Paper-Mask (30)	Lab-Dark, Lab-Lights, Sun *	[0.5, 1.5]	[−45, 45]	[−15, 15]
Spandex-Mask (30)	Lab-Dark, Lab-Lights, Sun *	[0.5, 1.5]	[−45, 45]	[−15, 15]
Face-Print Covid-Mask (30)	Lab-Dark, Lab-Lights, Sun *	[0.5, 1.5]	[−45, 45]	[−15, 15]

**Table 3 sensors-23-08914-t003:** Evaluating MobileNetV3 PAD algorithm noise sensitivity when using various training augmentations. The original dataset alone is the baseline control and designated by the *. The first comparison augmentation is geometric (designated by the ${}^{1}$). The second comparison augmentation is BSC (designated by the ${}^{2}$). Our proposed approach generates the best results, where the best combination is using original and noise-augmented training data (designated by the ${}^{†}$).

Train Dataset	Test Dataset	ACER	NPCER	APCER
Original ${}^{*}$	Original	0.9%	0.9%	0.9%
	Proposed Noises	13.7%	11.5%	16.0%
Geometric Augmented ${}^{1}$	Original	0.8%	0.7%	0.8%
	Proposed Noises	12.5%	12.1%	12.6%
BSC Augmented ${}^{2}$	Original	0.6%	0.7%	0.7%
	Proposed Noises	11.1%	9.7%	11.5%
Proposed Noises	Original	0.7%	0.4%	0.6%
	Proposed Noises	1.7%	0.5%	3.0%
Original and Proposed Noises ${}^{†}$	Original	0.7%	0.7%	0.7%
	Proposed Noises	1.0%	0.6%	1.4%

**Table 4 sensors-23-08914-t004:** Evaluating the use of proposed noise augmentations as data replacement. This table uses the same MobileNetV3 PAD algorithm for a data ablation study, where the original fully contrastive dataset is intentionally degraded. The first subset uses no participant-liveliness contrast (indicated by the ${}^{--}$); i.e., each person only has either live or spoof imagery (not both). The second subset uses partial participant-liveliness contrast (indicated by the ${}^{-}$); i.e., 30% of the participants are fully contrastive, but the remaining 70% have only live or spoof imagery. While the performance is achieved using the propose augmentations on the fully contrastive data (given by the ${}^{†}$), we demonstrate that this technique enables partially contrastive data (given by the ${}^{‡}$) to outperform the original (given by the *).

Train Dataset	Test Dataset	ACER	NPCER	APCER
**No Participant-Liveliness Contrast**				
Original ${}^{--}$	Original	13.2%	20.8%	5.7%
Original ${}^{--}$	Proposed Noises	22.2%	20.0%	24.4%
Proposed Noises ${}^{--}$	Original	13.6%	15.6%	11.5%
Proposed Noises ${}^{--}$	Proposed Noises	6.4%	9.5%	3.4%
Clean and Proposed Noises ${}^{--}$	Original	3.6%	4.0%	3.2%
Clean and Proposed Noises ${}^{--}$	Proposed Noises	4.5%	2.8%	6.1%
**Partial Participant-Liveliness Contrast**				
Original ${}^{-}$	Original	2.6%	1.2%	4.0%
Original ${}^{-}$	Proposed Noises	13.7%	11.5%	16.0%
Proposed Noises ${}^{-}$	Original	5.6%	6.3%	4.9%
Proposed Noises ${}^{-}$	Proposed Noises	5.1%	0.2%	10.0%
Original and Proposed Noises ${}^{-‡}$	Original	0.8%	1.0%	0.6%
Original and Proposed Noises ${}^{-‡}$	Proposed Noises	1.2%	1.2%	1.2%
**Full Participant-Liveliness Contrast (Original)**				
Original ${}^{*}$	Original	0.9%	0.9%	0.9%
Original ${}^{*}$	Proposed Noises	13.7%	11.5%	16.0%
Proposed Noises	Original	0.7%	0.4%	0.6%
Proposed Noises	Proposed Noises	1.7%	0.5%	3.0%
Original and Proposed Noises ${}^{†}$	Original	0.7%	0.7%	0.7%
Original and Proposed Noises ${}^{†}$	Proposed Noises	1.0%	0.6%	1.4%

## Data Availability

Full dataset is not available due to IBR requiring facial images to be stored locally. Images containing just the authors can be provided upon request.

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
