# Peer review of "Improving Monocular Facial Presentation–Attack–Detection Robustness with Synthetic Noise Augmentations"

_sensors, 2023, doi:10.3390/s23218914_

Round 1

Reviewer 1 Report

Comments and Suggestions for Authors
  1. The abstract section is clear enough;
  2. The related work section is well written.
  3. The experimental results sustain the drawn conclusions.
  4. The manuscript presentation is good.
Comments on the Quality of English Language

Some formulations could be improved.

Author Response

Dear reviewer,

Thank you for taking the time to read our paper and provide feedback. We found the comments helpful and made notable improvements towards the paper's clarity.

Please see the attachment for our commentary response.

Reviewer 2 Report

Comments and Suggestions for Authors

This paper presents a method to improve monocular facial presentation-attack-detection robustness with synthetic noise augmentations.

The studied topic is meaningful.

The authors are suggested to improve the paper from the follow aspects.

First, the writing and presenting of the paper still need improvement.

Some writing, formatting, and language issues are still observed.

As stated by the authors “… Real-world scenarios can introduce noises that effect image quality, degrading sharpness and contrast… ”, image quality is as important aspect for monocular face presentation-attack-detection. Some related surveys and works are suggested to be given for better understanding of the related topics, e.g., Screen content quality assessment: overview, benchmark, and beyond; Automatic prediction of perceptual image and video quality

One core strategy of this study is to use synthetic data to enhance the model ability in practical situations. This strategy has been widely used in various research areas.

For example the dehazing quality evaluation discussed in Image quality assessment: from error visibility to structural similarity; No-reference image quality assessment in the spatial domain. The authors are suggested to give some introductions on this aspect and the above works.

Section 3 Physics-Informed Noise-Augmentations Methodology

During the modeling of real-word noises, the authors generate several types of distortions for the data. Many distortions have been discussed and simulated in the research field of image quality assessment.

Visual attention can be of great value in various computer vision applications, and incorporating visual attention could improve these systems, for example is it possible to incorporate visual attention in this study?

All reference items are suggested to be double-checked. The formats should be unified.

Comments on the Quality of English Language

None

Author Response

(The authors gave the same response as above.)

Reviewer 3 Report

Comments and Suggestions for Authors

Please review the file.

Comments on the Quality of English Language

It is better to write articles in the third person.

Author Response

(The authors gave the same response as above.)

Reviewer 4 Report

Comments and Suggestions for Authors

Clarity of Problem Statement: The paper should provide a more explicit and concise problem statement at the beginning to clearly define the challenge being addressed. Specify the limitations of existing monocular face presentation attack-detection (PAD) algorithms and why robustness to real-world noise additions is crucial.

Methodology Explanation: Enhance the clarity of the methodology section. Provide a more detailed explanation of how the proposed physics-informed noise-augmentation approach works. Explain the rationale behind selecting the twelve sensor and lighting effect generators.

Baseline Comparison: While you mention that the toolbox generates more robust PAD features than popular augmentation methods, consider providing a thorough comparison with baseline methods and state-of-the-art approaches. This will help establish the significance of your proposed approach.

Data Collection Strategy: Describe the data collection strategy in more detail. Explain how the training data was collected, including the conditions under which the real-world noise was captured, to give readers a better understanding of the dataset's characteristics.

Experimental Design and Statistical Analysis: Provide more information on the experimental design, including the evaluation metrics used and the statistical significance of the results. Readers should have a clear understanding of how the experiments were conducted and how the conclusions were drawn.

Discussion of Practical Implications: Discuss the practical implications and potential applications of your findings. How might this improved PAD approach be applied in real-world authentication systems, and what are the potential benefits and limitations? This can help readers understand the broader impact of your work.

Please avoid citing sources that were published before to 2019. Cite current research that are really pertinent to your topic. The study also lacks sufficient citations. Another critical step is to compare the topic of the article to other relevant recent publications or works in order to widen the research's repercussions beyond the issue. Authors can use and depend on these essential works while addressing the topic of their paper and current issues.

A. Heidari, N. Jafari Navimipour and M. Unal, "A Secure Intrusion Detection Platform Using Blockchain and Radial Basis Function Neural Networks for Internet of Drones," in IEEE Internet of Things Journal, vol. 10, no. 10, pp. 8445-8454, 15 May15, 2023, doi: 10.1109/JIOT.2023.3237661.

Aldhyani, Theyazn HH, and Hasan Alkahtani. "Artificial Intelligence Algorithm-Based Economic Denial of Sustainability Attack Detection Systems: Cloud Computing Environments." Sensors 22.13 (2022): 4685.

Aldhaheri, Sahar, and Abeer Alhuzali. "SGAN-IDS: Self-Attention-Based Generative Adversarial Network against Intrusion Detection Systems." Sensors 23.18 (2023): 7796.

Comments on the Quality of English Language

 Minor editing of English language required

Author Response

(The authors gave the same response as above.)

Round 2

Reviewer 2 Report

Comments and Suggestions for Authors

None

Comments on the Quality of English Language

None

Reviewer 4 Report

Comments and Suggestions for Authors

No more comments.

Comments on the Quality of English Language

Minor editing of English language required